# Multicriteria Approach for Design Optimization of Lightweight Piezoelectric Energy Harvesters Subjected to Stress Constraints

Georgia Foutsitzi [1,*], Christos Gogos [1], Nikolaos Antoniadis [1] and Aris Magklaras [2]

[1] Department of Informatics and Telecommunications, University of Ioannina, GR 47100 Arta, Greece; cgogos@uoi.gr (C.G.); nadon@uoi.gr (N.A.)
[2] Department of Electrical and Computer Engineering, University of Patras, GR 26500 Patras, Greece; a.magklaras@upnet.gr
* Correspondence: gfoutsi@uoi.gr

**Abstract:** In this work a multicriteria optimization approach to minimize weight and maximize power output in piezoelectric energy harvesting systems for aerospace applications is studied. The design variables are the geometric and electric circuit parameters of the vibration-based piezoelectric energy harvester (PEH). A finite element model is developed to model the dynamic behavior of the composite plate-type harvester with embedded piezoelectric layers. The cantilever PEH structure is subjected to constraints in the bending stresses which must be lower than or equal to the tensile yield strength of the piezoelectric material. For solving the multi-objective optimization problem, the Non-dominated Sorting Genetic Algorithm II (NSGA-II), the Non-dominated Sorting Genetic Algorithm III (NSGA-III) and the Generalized Differential Evolution 3 (GDE3) algorithm are employed. It is shown that the proposed algorithms are effective in developing optimal Pareto front curves for maximum electrical power output and minimum mass of the PEH system. A comparative assessment of the solutions generated on the Pareto Front show that GDE3 achieved solutions that generate higher maximum power output and performs better compared to the two other algorithms.

**Keywords:** piezoelectric energy harvesting; stress constraint; unmanned aerial vehicles (UAV); multi-objective optimization; NSGA-II; GDE3; performance metrics; PlatEMO

## 1. Introduction

Energy harvesting (EH) from environmental energy sources such as vibrations, wind, heat, solar power etc., has attracted significant research interest due to the growing demand for energy. Vibration energy harvesting is the conversion of the ambient vibrations into electrical energy and provides a viable solution for powering small electronic components. Among the principles of energy conversion, piezoelectricity is known as one of the most efficient and practical way for conversion of mechanical vibration energy into electrical energy [1,2].

Piezoelectric energy harvesters (PEHs) have attracted research interest because of the high conversion efficiency compared to electromagnetic and electrostatic based harvesters [2]. However, efficiency of PEH systems depends on several parameters such as material properties, geometric dimensions, electric circuit components, etc. Conventional piezoelectric materials, such as lead-based piezoceramics, are brittle in nature and difficult to manufacture. Despite the high electromechanical response of lead-based piezoelectric ceramics, the problem of brittle nature has not been effectively solved for a long time. To overcome these drawbacks, new flexible piezoelectric composite materials have attracted attention of the research interest [3]. These new type of piezoelectrics are suitable for wearable energy harvester applications. Wearable energy harvesting focuses on the fabrication of reasonable-cost smart garments using special nanostructure piezoelectric fibers that make use of the vibrations due to the natural movements of the body [4,5]. Various materials are proposed, e.g., barium titanate and polyvinylidene fluoride, with

satisfactory results, indicating the potential for industrial mass production of commercial devices. However, this approach seems more appropriate for personalized sensing and monitoring applications in everyday life, such as patient monitoring, robotic assistance, etc.

Piezoelectric energy harvesters are typically designed as cantilever beams or plates with one or two piezoceramic layers covering the structure either entirely or partially. Several methods have been employed to model the electromechanical behavior of a PEH. The works of Erturk and Inman [6,7] present analytical distributed electromechanical models for unimorph and bimorph cantilever beam which provides closed form expressions for harmonic behavior of PEHs. Based on classical laminated theory, a distributed parameter electroelastic model was developed in [8] for piezoelectric energy harvester structurally integrated to cantilever composite beam. Electrical and mechanical closed form steady state solution response have been obtained by harmonic base excitation.

On the other hand, the finite element method has proven to be very useful in modeling the dynamics of PEHs [9–11]. A coupled electromechanical finite element (FE) model for predicting the electrical power output of piezoelectric energy harvester plates was presented in [9]. The FE formulation is based on the Kirchhoff plate assumptions which is suitable for modelling thin structures. Additionally in this paper, an optimization problem for aluminum wing spar generator of an unmanned air vehicle (UAV) was solved for the maximum electrical power without exceeding a prescribed mass addition limit.

Most analytical and FE models of PEHs are based on classical beam/plate theories which ignore shear stresses and are suitable for modelling thin structures. However, for accurate modeling of thick PEH for various applications, such as aircraft wing structure or wind turbine blade, higher order shear deformation theories are needed. Recently, Khazaee et al. [11] developed a coupled electromechanical model for non-uniform piezoelectric energy harvesting composite laminates based on third-order shear deformation theory. The presented high-order shear FE model also considers the contact layer thickness in the harvester beams, non-uniformity in the piezoelectric sheet, non-constant thickness of the piezoelectric sheet and is suitable for analysis of a wider range of problems in piezoelectric harvesting.

On the other hand, several studies have been carried out on design optimization of PE harvester to improve the energy harvesting efficiency by optimizing the dimensions of the piezoelectric energy harvesters [10,12,13]. The performance of few important piezoelectric materials has been simulated by Kumar et al. [10] for unimorph-type cantilever piezoelectric energy harvester. The genetic algorithm (GA) optimization approach is used to optimize the structural parameters of mechanical energy-based energy harvester for maximum power density. In [12], a new design of piezoelectric energy harvester subject to tip excitation is proposed. The mechanical and electrical behaviors of piezoelectric materials are solved by coupled analysis using ANSYS, and the design optimization is performed for power maximization using Sequential Quadratic Programming (SQP) algorithm.

Most studies on design optimization of PEH are limited to single objective optimization techniques using the maximization of power output as the main performance criterion. However, not much research has been carried out on the optimization of the parameters of the vibration-based piezoelectric harvester based on multicriteria.

The concept of energy harvesting has received much attention in recent years to enhance operational autonomy of low-power electronic applications (biosensors, micro-electronics etc.) as well as for aerospace applications (e.g., unmanned aerial vehicle (UAV)), as it can offer a sustainable solution for power supply from ambient vibrations. However, a crucial aspect of the design of such kind of systems is the potential effect that the additional mass of the piezoelectric energy harvesting system might have on the performance of the initial structure. Since mass densities of typical piezoceramics used in energy harvesting are considerably large compared to typical substrate materials such as steel, aluminum or graphite/epoxy material, the minimization of the mass of the system should consider as an additional performance criterion in optimal design of PEH.

A design optimization problem for UAV applications has been studied in [9]. The aluminum wing spar of a UAV is modeled using a FE plate model appropriately modified to design a generator wing spar. In order to take into consideration the mass added by the piezoelectric layers, an upper limit for mass addition is imposed as a design constraint. The resistor load *R* and the geometric dimensions of the embedded piezoceramics have been determined to maximize the generator spar's output power by varying their values in a reasonable range without applying any optimizationtechnique.

Recently, a multi-objective design optimization of piezoelectric energy harvesting system for UAV has been presented in [14]. In contrast with the previous approaches, this work considers the minimization of mass added by the embedded piezoceramics as an additional performance criterion along with the maximization of the power output as design optimization objectives. Non-dominated Sorting Genetic Algorithm II (NSGA-II), Non-dominated Sorting Genetic Algorithm III (NSGA-III) and Generalized Differential Evolution 3 (GDE3) algorithms are carried out to optimize the structural and the electric circuit parameters of vibration-based piezoelectric energy harvester. The results prove that Multi-Objective Genetic Algorithm (MOGA) approach is very promising for optimal design of PEH for aerospace applications.

So far, the literature review shows that the consideration of the material strength in optimal design of PEH is somewhat limited. In [12], the design of piezoelectric energy harvester subject to tip excitation is addressed under the constraint of maximum bending stress. The work of [15] focuses on nonlinear energy harvester design optimization with magnetic oscillator under the constraint that the maximum strain on piezoelectric material do exceed the allowable limits. However, the strength of the piezoelectric material is another crucial parameter in designing energy harvesters. This parameter is of major importance since the values of the strength of piezoelectric materials are much lower compared to the strength of substrate materials such as steel, aluminum, and brass [9,15]. Therefore, the stress generated in the energy harvesting process should be considered as a new design constraint in order to ensure the adequate mechanical behavior of the device.

Motivated by the above consideration, this study presents a multicriteria optimization approach to minimize mass and maximize power output in piezoelectric energy harvesting systems within the limits of allowable stress of the piezoelectric layers. A finite element model has been developed for modeling the behavior of the plate-type PEH under base excitation. The formulation is based on laminated plate theory combined with the first-order shear deformation theory (FSDT) for which each piezoelectric layer has one additional electrical degree of freedom. This paper extends the modeling and the optimization problem presented in [14] by considering additional constraint on bending stress of the piezoelectric layers. NSGA-II, NSGA-III and GDE3 algorithms are applied in the optimization process and both trade-off Pareto optimal fronts and the respective optimal design are obtained. Finally, the results are analyzed and discussed.

## 2. Multi-Objective Genetic Algorithms

Since the 1970s, multi-objective optimization problems (MOOP) have attracted the interest of both academics and industry. However, most of the time, they were treated as single-objective problems, aggregating multiple objectives in one using a weighted sum. The goal of single-objective optimization is to find the best solution, which is the minimization or maximization of a single objective function. Multi-objective optimization problems, on the other hand, usually, do not have a single optimal solution, but rather a set of compromise alternatives known as Pareto optimal solutions. So, the purpose of multi-objective optimization is to find solutions that yield the best values for more than one, often conflicting to each other, objectives.

Developments in technology and in heuristic approaches, particularly evolutionary algorithms, have enabled the development of current optimization tools that can solve multi-objective problems efficiently and reliably. Evolutionary algorithms have been used to solve a variety of multi-objective optimization problems, including economics (e.g.,

the economic dispatch problem [16], finance [17], optimal control [18], scheduling [19], and others).

In this research, we compare three state-of-the-art multi-objective genetic algorithms, NSGA-II, NSGA-III, and GDE3, to solve a multi-objective constrained optimization problem involving piezoelectric energy harvesting systems for aerospace applications.

## 2.1. NSGA-II and NSGA-III

Since NSGA-III expands on the core ideas of NSGA-II, the NSGA-II and NSGA-III algorithms are briefly discussed simultaneously. The NSGA-II algorithm employs both an elite and a diversity-preserving mechanism to achieve two objectives: discover solutions that are as close to the Pareto optimal as possible and find solutions that are as diverse as possible in the produced non-dominated front [20,21]. The initial population is sorted into distinct non-dominance levels. Based on the fitness of each solution, the solution is assigned to a previously determined non-domination level. The development of the offspring population for next steps, is dependent on selection mechanisms. A niching technique is used to choose a diverse collection of solutions, followed by crowded tournament selection, crossover, and mutation operations that ensure better distribution of the solutions.

NSGA-III, on the other hand, determines a set of reference points before generating the initial population. During the process, the solutions indicated by the Pareto front, are associated with these reference points. In essence, the reference point set serves as a guidance mechanism for the evolution towards creating a uniform Pareto front in the objective space [22].

## 2.2. GDE3

GDE3 is a global optimization approach that is based on the standard Differential Evolution (DE) method and can solve problems with many constraints and objectives [23]. The selection method differentiates DE and GDE from an algorithmic standpoint. In the initial versions of GDE, the trial vector was chosen to replace the old vector in the following generation if it weakly constraint-dominated the old vector by changing the selection rule of the basic DE. GDE3, extends its predecessor GDE2 [23], which can choose between achievable and non-dominating aims based on crowdedness in specific conditions, in that it can solve problems with multiple objectives and multiple constraint functions by extending the DE/rand/1/bin method [24]. So, decisions are based on the objective values and crowdedness while ensuring feasibility. A strong feature of GDE3 is that it appears to be less reliant on the selections of control parameters than other MOGAs.

## 3. Mathematical Model of a Piezoelectric Energy Harvester

The configuration of the structure studied is depicted in Figure 1. A piezoelectric energy harvester (PEH) consists of a host plate made of metal or composite materials with piezoelectric layers covering both surfaces in a bimorph arrangement. The piezoelectric layers need not extend along the entire length of the structure. The bond between two layers is assumed to be perfect and the physical properties of the bonding material are not considered. The piezoelectric layers are poled in the thickness direction and are covered by continuous electrodes with negligible thickness. Top and bottom electrodes of piezoelectric layers are connected to the external resistance $R$ as shown in Figure 1. The piezoelectric harvester is constrained by the clamped-free boundary conditions, and it is excited by the motion of its base.

Next, first order shear deformation theory (FSDT) is employed to derive a finite element model for the composite plate structure shown in Figure 1. Single mechanical displacement field is considered for all layers (equivalent single layer theory), while electric fields are considered for each piezoelectric layer separately (layerwise approach).

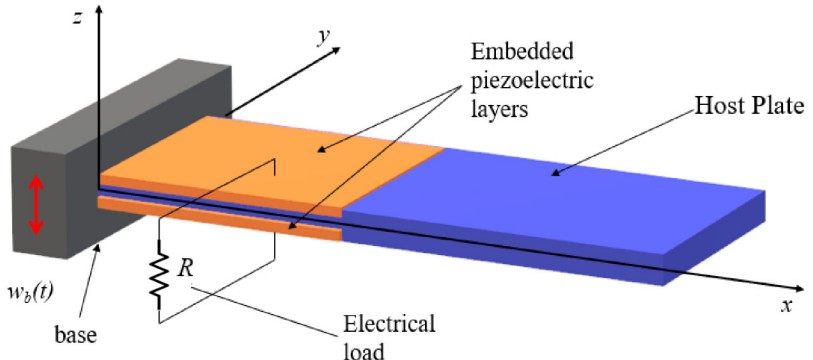

**Figure 1.** A cantilever bimorph PEH harvester with a load resistance in series connection.

### 3.1. Mechanical Displacements and Strains

Based on FSDT, the displacement field can be expressed as follow,

$$\{u\} = \left\{ \begin{array}{c} u_x(x,y,z,t) \\ u_y(x,y,z,t) \\ u_z(x,y,z,t) \end{array} \right\} = \left\{ \begin{array}{c} -z\psi_x(x,y,t) \\ -z\psi_y(x,y,t) \\ w_0(x,y,t) \end{array} \right\} = \left[ \begin{array}{ccc} 0 & -z & 0 \\ 0 & 0 & -z \\ 1 & 0 & 0 \end{array} \right] \left\{ \begin{array}{c} w_0 \\ \psi_x \\ \psi_y \end{array} \right\} \equiv [Z_u]\{\overline{u}\} \quad (1)$$

where $u_x$, $u_y$, $u_z$ are the displacement components along the $(x, y, z)$ coordinates and $\{\overline{u}\} = \{w, \psi_x, \psi_y\}^T$ indicates the transverse displacement and the section rotations of the mid-surface of the plate. Using the usual strain–displacement relations in conjunction with relations (1), the strain field can be written as,

$$\{\varepsilon\} = \left\{ \begin{array}{c} \varepsilon_{xx} \\ \varepsilon_{yy} \\ \gamma_{xy} \\ \gamma_{yz} \\ \gamma_{xz} \end{array} \right\} = \left\{ \begin{array}{c} -z\frac{\partial \psi_x}{\partial x} \\ -z\frac{\partial \psi_y}{\partial y} \\ -z\left(\frac{\partial \psi_x}{\partial y} + \frac{\partial \psi_y}{\partial x}\right) \\ \frac{\partial w}{\partial y} - \psi_y \\ \frac{\partial w}{\partial x} - \psi_x \end{array} \right\} = [Z_\varepsilon][\nabla_u]\{\overline{u}\} \quad (2)$$

where,

$$[Z_\varepsilon] = \left[ \begin{array}{ccccc} -z & 0 & 0 & 0 & 0 \\ 0 & -z & 0 & 0 & 0 \\ 0 & 0 & -z & 0 & 0 \\ 0 & 0 & 0 & 1 & 0 \\ 0 & 0 & 0 & 0 & 1 \end{array} \right], \ [\nabla_u] = \left[ \begin{array}{ccc} 0 & \frac{\partial}{\partial x} & 0 \\ 0 & 0 & \frac{\partial}{\partial y} \\ 0 & \frac{\partial}{\partial y} & \frac{\partial}{\partial x} \\ \frac{\partial}{\partial y} & 0 & -1 \\ \frac{\partial}{\partial x} & -1 & 0 \end{array} \right] \quad (3)$$

### 3.2. Constitutive Relations

In this work the linear constitutive equations of piezoelectricity are employed,

$$\{\sigma\}_p = [Q]_p\{\varepsilon\} - [e]_p^T\{E\}_p, \ \{D\}_p = [e]_p\{\varepsilon\} + [\xi]_p\{E\}_p \quad (4)$$

where $\{\sigma\}_p$, $\{\varepsilon\}$, $(D)_p$, $\{E\}_p$ are stress, strain, electric displacement and electric field vector, respectively. $[Q]_p$, $[e]_p$ and $[\xi]_p$ are plane-stress reduced stiffness matrix, the piezoelectric coefficients and the permittivity constant matrices, respectively. Matrix transposition is denoted by superscript $T$. The electric field vector $\{E\}_p$ of the $p-th$ piezoelectric layer can be derived from the electric potential $\phi_p$ as,

$$\{E\}_p = -\nabla \phi_p \quad (5)$$

Relation (4) can be greatly simplified since piezoelectric materials are transversally isotropic in the plane normal to the axis of polarization $z$ (see, e.g., [25]).

The constitutive relation for the elastic substructure material can be written as,

$$\{\sigma\}_S = [Q]_S\{\varepsilon\}. \tag{6}$$

### 3.3. Finite Element Discretization

In order to derive the coupled electromechanical equations governing the dynamic behavior of the PEH, the finite element method is used. The essential idea of finite elements is that the structure is approximated as an assembly of elements linked at nodal points on the element boundaries. In this study, the overall structure has been discretized using four-noded isoparametric quadrilateral elements with three mechanical degrees of freedom (DoF) per node and one electrical degree of freedom per piezoelectric layer. The generalized displacement vector is discretized on a quadrilateral element as:

$$\{\bar{u}(x,y,t)\}_e = [N_u(x,y)]\{d^e(t)\} = \sum_{j=1}^{4} (N_j[I]_{3x3}\{d_j^e\}) \tag{7}$$

where $\{d_j^e\} = \{w_{0j}, \psi_{xj}, \psi_{yj}\},^T j = 1,2,3,4$ corresponds to the $j^{\text{th}}$ node of the $e^{th}$ element and $N_j(x,y)$ are the linear Lagrange interpolation functions.

Substituting Equation (7) into Equation (2), the strain vector at any point within an element can be expressed as,

$$\{\varepsilon(x,y,t)\}_e = [Z_\varepsilon]\nabla_u([N_u]\{d^e\}) = [Z_\varepsilon][B]\{d^e\}. \tag{8}$$

For a thin piezoelectric layer polarized in the thickness direction, the electrical potential can be assumed to be constant throughout the plane of the element and to vary linearly along the $z$-direction. Therefore, the electric field strengths of an element for the lower ($p_1$) and upper ($p_2$) piezoelectric layer can be accurately approximated as,

$$\{E^e\}_{p1} = \left\{0 \; 0 \; -\frac{v_1}{h_{p_1}}\right\}_e^T = -[B_{p1}]v_1^e \tag{9}$$

$$\{E^e\}_{p2} = \left\{0 \; 0 \; -\frac{v_2}{h_{p_2}}\right\}_e^T = -[B_{p2}]v_2^e \tag{10}$$

where $v_i^e$ is the difference of electric potential between the electrodes covering the surface on each side of the piezoelectric layer $i$. Notice that in this way, two additional electric DoFs per element have been included, namely the electric potential difference $v_1^e$ and $v_2^e$ at the top of the lower and upper piezoelectric layers.

### 3.4. Variational Formulation

The equations of motion for the energy harvester composed of elastic and piezoelectric layers are obtained using the extended Hamilton's principle,

$$\delta \int_{t_1}^{t_2} (T - (U_m + U_E) + W)dt = 0 \tag{11}$$

where $t_1$ and $t_2$ are arbitrary time moments, $T$ is the mechanical kinetic energy, $U_m$ is the mechanical potential energy $U_E$ is the electrical potential energy, $W$ is the external work and $\delta$ denotes the variational operator.

In the Finite Element Method, the structure is divided into a finite number of elements and therefore, the various energy terms must be computed for each element and subsequently to be assembled all together in order to describe the whole structure. We emphasize

that each finite element of the mesh is either a single elastic layer, when only the substrate is present, or a multilayer, when both the substrate and the piezoelectric layers are present. Next, all energy terms are given for a single multilayer element as follows,

$$T^e = \frac{1}{2} \int_{\Omega_S} \rho_S \{\dot{u}\}^T \{\dot{u}\} d\Omega + \frac{1}{2} \int_{\Omega_{p_1}} \rho_{p_1} \{\dot{u}\}^T \{\dot{u}\} d\Omega + \frac{1}{2} \int_{\Omega_{p_2}} \rho_{p_2} \{\dot{u}\}^T \{\dot{u}\} d\Omega \tag{12}$$

$$U_m^e = \frac{1}{2} \int_{\Omega_S} \{\varepsilon\}^T \{\sigma\}_S d\Omega + \frac{1}{2} \int_{\Omega_{p_1}} \{\varepsilon\}^T \{\sigma\}_{p_1} d\Omega + \frac{1}{2} \int_{\Omega_{p_2}} \{\varepsilon\}^T \{\sigma\}_{p_2} d\Omega \tag{13}$$

$$U_E^e = \frac{1}{2} \int_{\Omega_{p_1}} \{E\}_{p_1}^T \{D\}_{p_1} d\Omega + \frac{1}{2} \int_{\Omega_{p_2}} \{E\}_{p_2}^T \{D\}_{p_2} d\Omega \tag{14}$$

In the above equations, $\Omega$ denotes the volume, $\rho$ denotes the mass density and the subscripts $S$, $p_1$ and $p_2$ stand for the host plate structure, the lower and the upper piezoelectric layer, respectively, and a dot represents partial derivative with respect to time $t$.

Finally, the virtual work carried out by the mechanical forces and electrical charges for an element $e$, is given by,

$$\delta W^e = \{\delta u\}_L^T \{f^e\}_L + \delta v_1^e q_1^e + \delta v_2^e q_2^e \tag{15}$$

where $\{f^e\}_L$ is the applied mechanical force at $(x_L, y_L)$ position and $q_j^e$ is the charge extracted by the piezoelectric layer $j$.

Using the Equations (4)–(10) and (12)–(14) the Hamilton's principle (11) applied to an arbitrary piezoelectric element, can be expressed as,

$$\int_0^T \left\{ \delta \{d^e\}^T \left[ \int_{\Omega_s} [N]^T [Z_u]^T \rho_S [Z_u][N] d\Omega \left\{ \ddot{d}^e \right\} + \int_{\Omega_{p_1}} [N]^T [Z_u]^T \rho_{p1} [Z_u][N] d\Omega \left\{ \ddot{d}^e \right\} \right. \right.$$
$$+ \int_{\Omega_{p_2}} [N]^T [Z_u]^T \rho_{p_2} [Z_u][N] d\Omega \left\{ \ddot{d}^e \right\} + \int_{\Omega_s} [B]^T [Z_\varepsilon]^T [Q]_S [Z_\varepsilon][B] d\Omega \{d^e\}$$
$$+ \int_{\Omega_{p_1}} [B]^T [Z_\varepsilon]^T [Q]_{p_1} [Z_\varepsilon][B] d\Omega \{d^e\} + \int_{\Omega_{p_2}} [B]^T [Z_\varepsilon]^T [Q]_{p_2} [Z_\varepsilon][B] d\Omega \{d^e\}$$
$$+ \int_{\Omega_{p_1}} [B]^T [Z_\varepsilon]^T [e]_{p_1}^T [B_{v_1}] d\Omega \, v_1^e + \int_{\Omega_{p_2}} [B]^T [Z_\varepsilon]^T [e]_{p_2}^T [B_{v_2}] d\Omega \, v_2^e - \{f^e\} \right]$$
$$+ \delta v_1^e \left[ - \int_{\Omega_{p_1}} [B_{v_1}]^T [e]_{p_1} [Z_\varepsilon][B] d\Omega \{d^e\} + \int_{\Omega_{p_1}} [B_{v_1}]^T [\xi]_{p_1} [B_{v_1}] d\Omega \, v_1^e - q_1^e \right]$$
$$\left. + \delta v_2^e \left[ - \int_{\Omega_{p_2}} [B_{v_2}]^T [e]_{p_2} [Z_\varepsilon][B] d\Omega \{d^e\} + \int_{\Omega_{p_2}} [B_{v_2}]^T [\xi]_{p_2} [B_{v_2}] d\Omega \, v_2^e - q_2^e \right] \right\} dt = 0 \tag{16}$$

Since $\delta\{d^e\}$, $\delta v_1^e$ and $\delta v_2^e$ are independent and arbitrary, Equation (16) implies,

$$[M^e] \left\{ \ddot{d}^e \right\} + [K_u^e]\{d^e\} + [K_1^e]v_1^e + [K_2^e]v_2^e = \{f^e\} \tag{17}$$

$$- [K_1^e]^T \{d^e\} + C_{p1} v_1^e = q_1^e \tag{18}$$

$$- [K_2^e]^T \{d^e\} + C_{p2} v_2^e = q_2^e \tag{19}$$

where $[M^e]$, $[K_u^e]$, $[K_1^e]$, $[K_2^e]$, $C_{p1}$ and $C_{p2}$ are the element mass matrix, element stiffness matrix, electromechanical coupling matrices and piezoelectric capacitances, respectively. Their definitions follow directly by Equation (16).

In this work, the structure's motion is caused by the base's acceleration, and the effective force on the plate is caused by its inertia. As a result, the forcing term in the right-hand side of Equation (17) can be expressed as,

$$\{f^e\} = -[M^e]\ddot{w}_b \tag{20}$$

where $\ddot{w}_b = a_b\{1\,0\,0\,1\,0\,0\,1\,0\,0\,1\,0\,0\}^T$ and $a_b$ is the magnitude of the base acceleration. In our case, $a_b$ is taken as the gravitational acceleration, $g = 9.81$ m/s$^2$.

*3.5. Coupled Electromechanical Equations of the PEH*

The global coupled electromechanical equations of the PEH can be obtained by assembling the elemental Equations (17)–(19). To identify elements with piezoelectric layers, a unique numbering scheme should be utilized during the assembly procedure. Elements with piezoelectric layers, for example, may be denoted by 1, while the rest may be designated by 0. In addition, since the piezoelectric layers are fully covered by uniform electrodes, a single electrical degree of freedom $v_i$ for each piezoelectric layer is assigned, to consider the equipotential condition on the electrodes. Thus, the global coupled equations of the system are given as,

$$[M]\{\ddot{d}\} + [C]\{\dot{d}\} + [K_u]\{d\} + [\Theta_1]v_1 + [\Theta_2]v_2 = \{F_b\} \tag{21}$$

$$-[\Theta_1]^T\{d\} + C_{p1}v_1 = q_1 \tag{22}$$

$$-[\Theta_2]^T\{d\} + C_{p2}v_2 = q_2 \tag{23}$$

where $\{d\}$ is the global vector of mechanical coordinates, $[M]$ is the global mass matrix, $[K]$ is the global stiffness matrix, $\{F_b\}$ is the global force vector and $[\Theta_1]$, $[\Theta_2]$ are the electromechanical coupling matrices. $C_{pi}$ and $q_i$ are the capacitance and the electric charge output of the piezoelectric layer $i$, respectively. In Equation (21), a Rayleigh-type mechanical damping matrix $[C] = \alpha[M] + \beta[K]$ has been introduced a posteriori.

In order to use the model for energy harvesting applications, some additional considerations should be made. First, the two piezoelectric layers are considered identical (made of same material and have same dimensions) and poled in antiparallel directions. In this case, $C_{p1} = C_{p2}$. Secondly, in order to maximize the voltage output, the two opposite polarized piezoelectric layers have been connected in series to an external resistance $R$. In this case, the global charge in the circuit is equal to each output charge generated by each piezoelectric layer, i.e., $q_1 = q_2 = q$, whereas the global output voltage is the sum of the output voltages, i.e., $v = v_1 + v_2$. Summing up Equations (22) and (23), we obtain,

$$-([\Theta_1]^T + [\Theta_2]^T)\{d\} + C_{p1}v - 2q = 0. \tag{24}$$

Next, we differentiate the above equation and using Ohm's law ($I = \dot{q} = -v/R$), the complete system of equations governing the dynamic response of the PEH under study become:

$$[M]\{\ddot{d}\} + [C]\{\dot{d}\} + [K]\{d\} + [\Theta]v = \{F_b\} \tag{25}$$

$$-[\Theta]^T\{\dot{d}\} + C_p\dot{v} + \frac{v}{R} = 0 \tag{26}$$

where $C_p = \frac{C_{p1}}{2} = \frac{C_{p2}}{2}$ and $[\Theta] = \frac{1}{2}([\Theta_1] + [\Theta_2])$.

*3.6. Solution of the Coupled Electromechanical System*

In order to derive the electromechanical response of the PEH under base excitation, the system is assumed to be excited by a harmonic input of the form $\{F_b\} = \{F_0\}e^{-j\omega t}$. In this case, the steady-state solutions of the mechanical displacement and voltage responses can be formulated as $\{d\} = \{d_0\}e^{j\omega t}$ and $v = v_0e^{j\omega t}$ where $\{d_0\}$ and $v_0$ are the amplitudes of the displacement and the voltage, respectively, and $\omega$ is the driving frequency. Substituting

the assumed solutions in the system (25)–(26) and doing some mathematical manipulations, the mechanical displacement and the voltage output are found to be related to the input force by,

$$\{d_0\} = [H(\omega)]\{F_0\} \tag{27}$$

$$v_0 = \frac{j\omega}{(\frac{1}{R} + j\omega C_p)} [\Theta]^T [H(\omega)]\{F_0\} \tag{28}$$

In the above equations, the frequency response function (FRF) between the mechanical output and input signal is given by,

$$[H(\omega)] = \left( -\omega^2 [M] + j\omega [C] + [K] + \frac{j\omega}{(\frac{1}{R} + j\omega C_p)} [\Theta][\Theta]^T \right)^{-1} \tag{29}$$

Once the system's DoFs have been derived by Equations (27) and (28), the stresses can be evaluated using the constitutive equations.

The power FRF ($P$) is calculated using the voltage FRF and the load resistance as,

$$P = \frac{v^2}{R} \tag{30}$$

The modulus of the power output FRF will be used in the next section to define a multi-objective optimization problem.

A MATLAB code has been developed to implement the presented FE model, to perform the harmonic analysis and the post-processing computation of stresses.

### 3.7. Verification of the FE Model

The developed FE model has been verified by comparing with the experimental results of [7] for a bimorph cantilever harvester with a tip mass. The bimorph cantilever PEH consists of a brass substructure and sheets of piezoelectric material PZT-5A. The material and geometric parameters needed to construct the FE model are given in Table 1. The global mass matrix of the FE model has been appropriately modified at the tip degrees of freedoms in order to incorporate the additional mass. The experimental results have been recovered by the study [7] using the WebPlotDigitizer v4.5 software [26] which is a useful tool for accurate data extraction from plots and images.

**Table 1.** Material and geometric parameters of the bimorph PEH with tip mass.

| Parameter | Value |
| --- | --- |
| Beam length (mm) | 50.8 |
| Beam width (mm) | 31.8 |
| Substructure thickness (mm) | 0.14 |
| Piezoelectric thickness (mm) | 0.26 (each) |
| Substructure Young's modulus, $E_S$ [GPa] | 105 |
| Substructure Poisson's ratio, $v$ | 0.35 |
| Substructure density, $\rho_S$ $[\text{kg/m}^3]$ | 9000 |
| Rayleigh coefficient $\alpha$ [rad/s] | 14.65 |
| Rayleigh coefficient $\beta$ [s/rad] | $10^{-5}$ |
| Piezoelectric layer density, $\rho_p [\text{kg/m}^3]$ | 7800 |
| Tip mass [kg] | 0.012 |
| $Q_{11}^p = Q_{22}^p$ [GPa] | 70 |
| $Q_{12}^p$ [GPa] | 24.5 |
| $Q_{44}^p = Q_{55}^p$ [GPa] | 21.8 |
| $Q_{66}^p$ [GPa] | 22.8 |
| $e_{31}^p = e_{32}^p$ $[\text{C/m}^2]$ | 15.97 |
| $\tilde{\xi}_{11}^p = \tilde{\xi}_{33}^p$ [nF/m] | 11.0 |

The harmonic analysis of the PEH under base excitation has been performed using the developed MATLAB code. In Table 2 the fundamental frequency obtained by the present FE model is compared with the analytical and experimental frequencies in short-circuit and open-circuit condition obtained in [7] From the results of Table 2, an excellent agreement is obtained. Figure 2. FE model verification with the experimental results of [7]. Figure 2 depicts the power spectrum in an excitation range close to the first resonance frequency when a resistor $R = 1 \, \text{k}\Omega$ is connected in series with the piezoelectric layers. From Figure 2 it can be concluded that the predicted power output FRF obtained using the present FE model agrees excellent with the experimental results.

**Table 2.** The fundamental frequency (Hz) of the bimorph harvest with tip mass [7].

|  | **Short-Circuit** | **Open-Circuit** |
|---|---|---|
| Analytical | 45.7 | 48.2 |
| Experimental | 45.6 | 48.4 |
| Present FE | 45.7 | 48.3 |

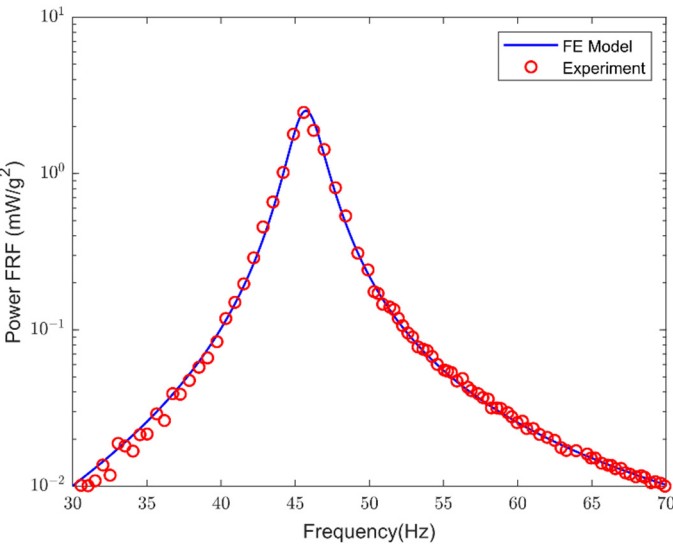

**Figure 2.** FE model verification with the experimental results of [7].

## 4. Design Optimization of PEH

In design optimization of PEH systems, the optimization formulation (objectives, constraints, method etc.) depends on the particular implementation requirements. For example, for applications where the mass of the system is of major concern (e.g., UVA applications), the designer should focus their attention not only on maximization of the power output but also on the competing objective of the mass minimization. Although the approach given here is limited to two specific objectives (generated power and PEH mass), it might be modified to include any additional objective and/or constrains.

Based on the FRF model developed in previous section, a design optimization of the geometric and electric parameters of PEHs subject to base excitation is studied next.

*Formulation of the Optimization Problem*

An optimization problem with multi-objectives is formulating in this section to design a cantilever plate-type PEH with embedded piezoceramics which can be used to simulate the dynamics of an UAV wing spar [14] The dimensions of the original PEH are given as $300 \times 30 \times 12 \, \text{mm}^3$. The standard material of the harvester is partly replaced by two PZT-5A layers that are embedded in both surfaces of the harvester (see Figure 1 for the configuration). The bimorph PEH consists of an aluminum substructure and the PZT-5A piezoceramic layers. The piezoelectric material constants are given in Table 1. The remain

physical properties of the bimorph PEH are listed in Table 3. The piezoelectric layers are assumed to have the same width as the substrate. The geometric properties of the piezoceramic layers, particularly their length and thickness, as well as the resistance of the electric circuit, are the design variables of the optimization problem The base excitation speed is assumed to be close to the short circuit resonance frequency of the first vibration mode of the system, that is in a range between 98 and 110 Hz. Recall that the base acceleration considered in this work is 1.0 g (9.81 m/s$^2$).

**Table 3.** Physical properties of the bimorph PEH.

| Parameter | Value |
|---|---|
| Substructure Young's modulus, $E_S$ [GPa] | 70 |
| Substructure Poisson's ratio, $v$ | 0.34 |
| Substructure density, $\rho_S$ [kg/m$^3$] | 2750 |
| Rayleigh coefficient $\alpha$ [rad/s] | 21.28 |
| Rayleigh coefficient $\beta$ [s/rad] | $10^{-5}$ |

Next, we define the dimensionless geometric parameters: the dimensionless length ($L^*$) which is equal to the ratio of the length of the piezoceramic layers ($L_p$) divided by the total length of the harvester ($L$) and the dimensionless height ($h^*$) which is equal to the ratio of the height of one piezoceramic layer ($h_p$) divided by the total height of the harvester (i.e., $L^* = L_p/L$ and $h^* = h_p/h$).

The main goal in vibration-based energy harvesting is the maximization of the power output $P$ to enhance the performance of the PEH system. So, the first objective considered in this study is the maximization of the peak power FRFs $P(\omega)$. Next, the maximization of the peak power FRFs is handle as a minimization of the following function,

$$f_1 = -\text{peak power}(L^*, h^*, R) = -\|P\|_\infty \tag{31}$$

where $\|P\|_\infty = \max_\omega |P(\omega)|$.

Since the mass density of PZT-5A is considerably larger than that of the aluminum alloy, the minimization of the mass of the PEH system is consider as an additional performance criterion in the following optimization problem:

$$f_2 = \text{mass}(L^*, h^*, R) = \Omega_S \left( \rho_s - 2L^* h^* \rho_s + 2L^* h^* \rho_p \right) \tag{32}$$

where $\Omega_S = Lbh$ is the volume of the original substrate structure.

Generally, the strength of piezoelectric materials is much lower compared to the strength of substrate materials such as aluminum considered in this study [9,15]. The reported value of dynamic strength of PZT-5A is 27.6 MPa [9]. This fact motivates us to incorporate the material strength of piezoelectric material in designing the PEH. Thus, in pursuit of better performance, the maximum bending stress is considered as a new constraint in the design optimization.

Thus, the multi-objective constrained optimization problem is formulated as follows: Find design variables $L^*, h^*, R$ to,

$$\begin{aligned}
\text{minimize} F(L^*, h^*, R) &\equiv (f_1, f_2) \\
st \quad 0 &\leq L^* \leq 1.0 \\
0 &\leq h^* \leq 0.5 \\
1 &\leq R \leq 600 \ (k\Omega) \\
\sigma_p &\leq \sigma_{PZT}^0
\end{aligned} \tag{33}$$

where $\sigma_p$ is the maximum bending stress of the piezoelectric layers and $\sigma_{PZT}^0 = 27.6$ (MPa) is the dynamic tensile strength of PZT-5A.

A multi-objective optimization problem with the same objectives and without the stress constraint has been solved in [14]. Furthermore, an additional constraint on the mass of the system was imposed that $f_2$ does not exceed the value 0.3267, which corresponds to 10% increase of the mass of the original structure. Depending on the particular implementation requirements, this constraint could be changed and it was chosen arbitrarily only to show the capabilities of this procedure to perform optimization under different constraints.

## 5. Optimization Results

Experiments were carried out on a workstation running MATLAB 2018b on Windows 10 with an Intel Core i9 7960X @2.8 GHz CPU and 64 GB DDR4 RAM. The three multi-objective algorithms employed are those that are implemented in the PlatEMO v3.4 [27] software, which is freely available for research purposes. It should be noted that PlatEMO implements numerous MOGA and other algorithms.

The optimization procedure is shown in Figure 3. The MATLAB code that we have developed for the FE model implementation is executed at each iteration to calculate the objectives and the constraints required by the optimization procedure (e.g., the output power, the mass, the stress distribution). PlatEMO software provides the implementations of the multi-objective algorithms used (NSGA-II, NSGA-III, GDE3). Our MATLAB code is embedded in the procedure as a set of functions, which, when the values for the decision variables are given as input, return the fitness of the objective functions and the constraints violation amount. The convergence criterion is a predefined number of maximum generations allowed. As long as the criterion is not satisfied, the algorithm continues to evolve to population. Finally, the Pareto optimal solutions are obtained. A post-processing procedure is carried out with the aid of two extra python packages, pfevaluator [28] that computes various Pareto front performance metrics, and OAPackage [29] that can easily identify Pareto optimal solutions in a population of solutions.

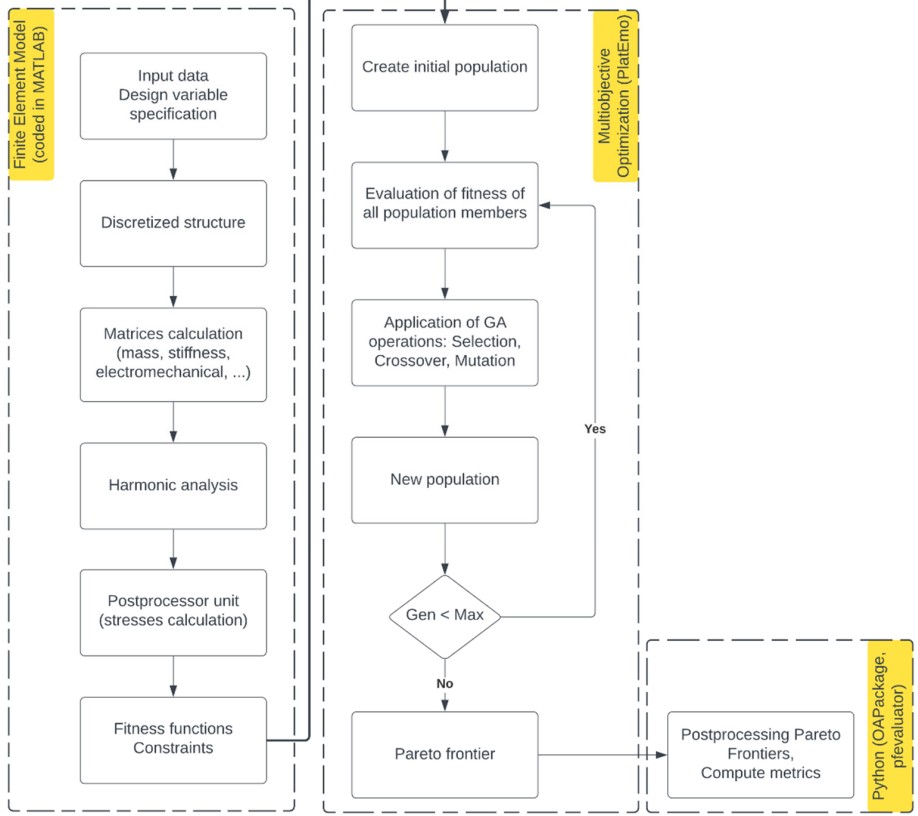

**Figure 3.** Flowchart of the optimization procedure.

The experiments entailed running each of the algorithms NSGA-II, NSGA-III, and GDE3 for 50 generations with a population of 50 individuals. Based on the collected findings of each run's final population, a Base Pareto Front (BPF) is created, as shown in Figure 4. The BPF comprises 493 points (135 from NSGA-II, 202 from NSGA-III, and 156 from GDE3). It is easily shown that GDE3 manages to extend the BPF to the top-right part of the graph. Each algorithm was run 10 times, each time taking roughly 5500 s. The indifference of the running times among the three algorithms can be attributed to the fact that the heavier parts of the approach, due to FEM, are the evaluation of the power objective and the assurance of the stress constraint which is common across the algorithms.

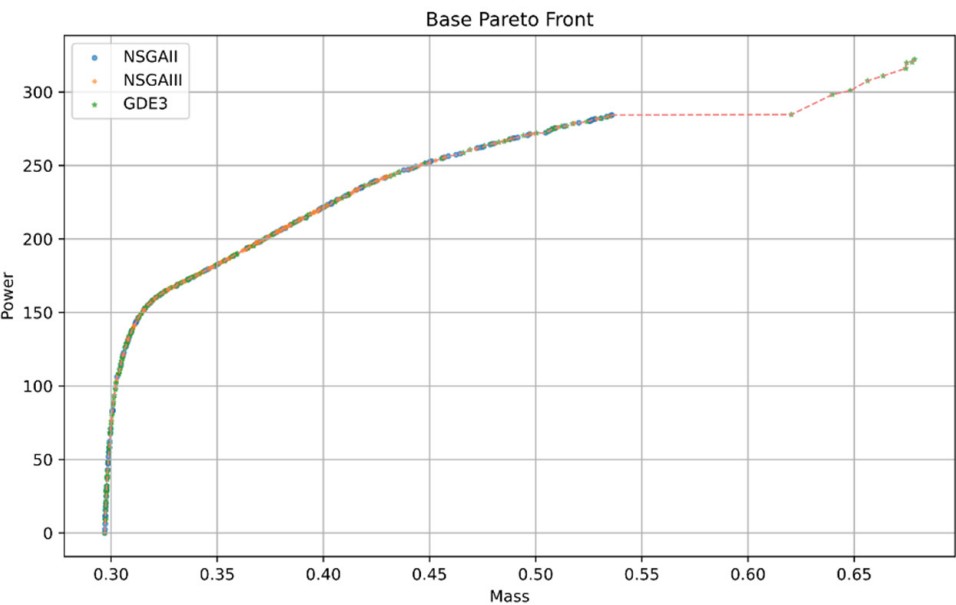

**Figure 4.** Base Pareto Front (BPF)—constructed from 10 runs of each algorithm (NSGA-II, NSGA-III, GDE3).

All three algorithms begin to converge, rather early, around the 10th generation. Figure 5 shows the evolution of the Pareto fronts for generations 3, 10, and 50. In addition, the Pareto fronts for generation 5, 10, and 50, are shown in Figure 6. BPF is also shown in each subfigure as a substitute for the unknown optimal Pareto front. GDE3 can achieve solutions that NSGA-II and NSGA-III were unable to attain for this problem and experiment setup. As a result, GDE3 achieves solutions that generate a maximum power of 322.2709 mW/g$^2$, whereas the maximum power generated by solutions produced by NSGA-II and NSGA-III is 284.3626 mW/g$^2$. By inspecting the evolution of solutions across the generations we observed that GDE3 attained solutions with power greater than 284.3626 mW/g$^2$ for all 10 runs and this occurred on average at about the 33th generation. This occurred as early as at the 18th generation and as late as at the 45th generation. More experiments were undertaken for NSGA-II and NSGA-III, in an attempt to achieve solutions with greater values for the generated power. A population of 100 individuals and 100 generations is used for the optimization process. This time, each run took about 21,000 s. Nevertheless, the maximum power generated is again no greater than 284.3626 mW/g$^2$.

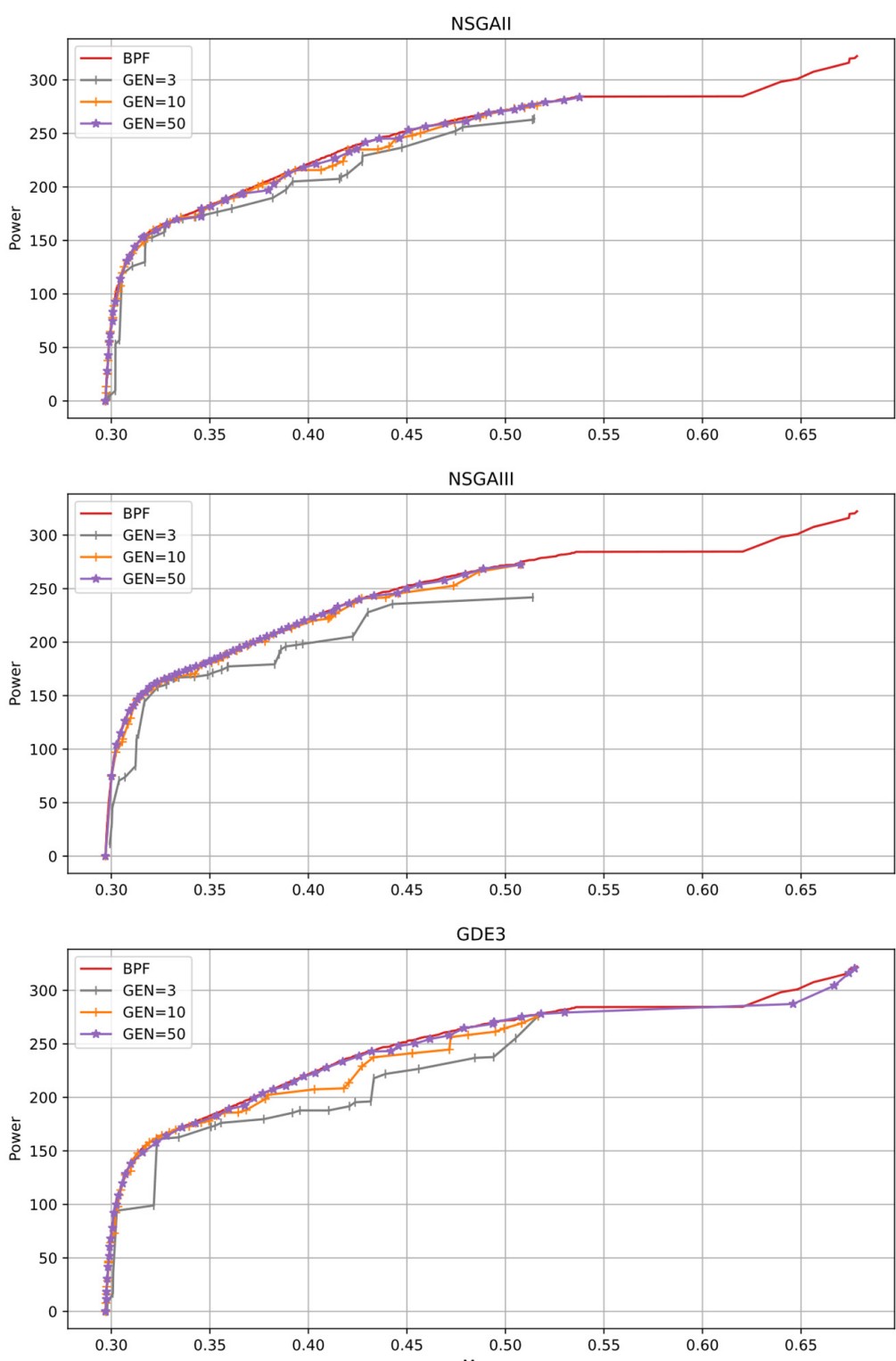

**Figure 5.** NSGA-II, NSGA-III, GDE3 Pareto front evolution through generations.

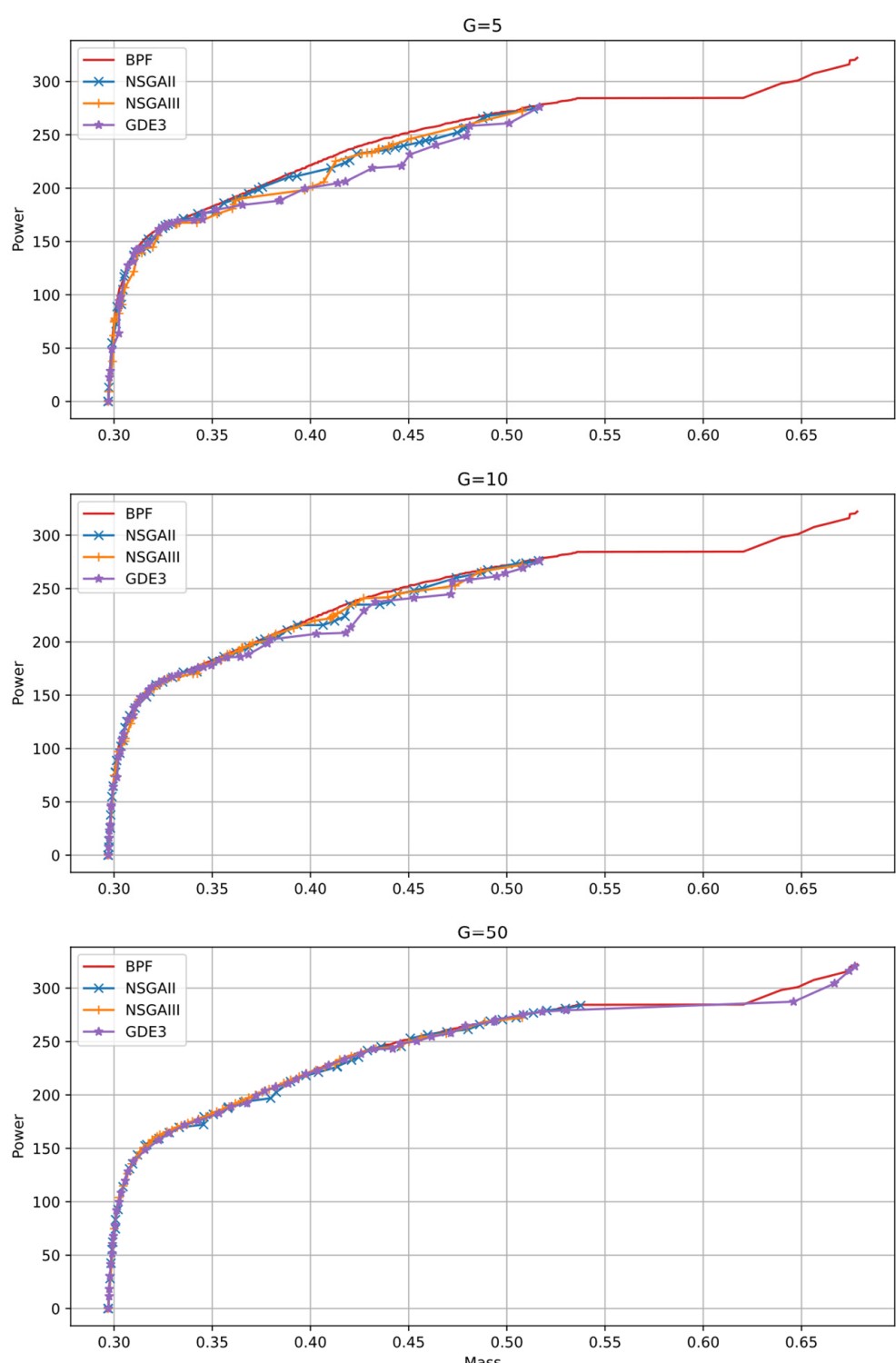

**Figure 6.** Pareto fronts produced by NSGA-II, NSGA-III and GDE3 at generations 5, 10 and 50.

### 5.1. Metrics

Several metrics exist in the literature that evaluate the performance of multi-objective algorithms, including Generational Distance (GD), Inverted Generational Distance (IGD), Maximum Pareto Front Error (MPFE), Hypervolume (HV), Maximum Spread (MS), and others [30]. The optimal Pareto front is used to calculate most of the metrics. We employed the Base Reference Front (BPF), which comprises the Pareto optimal solutions of 10 runs of

the algorithms NSGA-II, NSGA-III, and GDE3, because we had no theoretical guarantee of the best Pareto front.

The GD metric evaluates the distance between the solution Pareto front and the BPF, with smaller values signifying better performance. It calculates the average distance between any place in the BPF and the solution Pareto front's nearest point. Since the IGD metric is an inverted version of the GD, higher values are preferable. IGD specifically assesses how well the Pareto Front solution represents the BPF. The maximum distance between a point in the solution Pareto front and the nearest point in the BPF is measured by the MPFE metric. Smaller numbers are preferred because MPFE represents mistake. HV is computed against a reference point, rather than a reference front, that is dominated by all solutions of the solution Pareto front, and measures both closeness and diversity. Larger numbers are preferred for HV. Finally, MS indicates how well the solution Pareto front covers the BPF, with higher MS values indicating better coverage.

In Table 4 metric values that were computed over 10 runs of each algorithm against the BPF are presented. Best values across algorithms are indicated by bold style. GDE3 performs best according to MPFE, MS and HV metrics, since it manages to extend the Pareto front to values of power greater than those of the two other algorithms.

**Table 4.** Performance Metrics for NSGA-II, NSGA-III and GDE3 (Population = 50, Generations = 50).

|          |      | GD         | IGD        | MPFE        | MS         | HV            |
|----------|------|------------|------------|-------------|------------|---------------|
| NSGA-II  | Mean | 0.0863     | 0.2380     | 8.3296      | 0.7398     | 38,026.42     |
|          | SD   | 0.0073     | 0.0508     | 2.5134      | 0.0349     | 722.97        |
|          | Best | 0.0737     | 0.3158     | 5.3478      | 0.7652     | 38,566.79     |
| NSGA-III | Mean | **0.0576** | **0.4174** | 35.4863     | 0.7307     | 37,782.24     |
|          | SD   | 0.0067     | 0.0223     | 2.5286      | 0.0264     | 600.21        |
|          | Best | 0.0518     | 0.4405     | 32.3415     | 0.7623     | 38,477.20     |
| GDE3     | Mean | 0.0870     | 0.1121     | **6.3600**  | **0.9743** | **41,234.90** |
|          | SD   | 0.0124     | 0.0265     | 1.2441      | 0.0295     | 1227.58       |
|          | Best | 0.0740     | 0.1769     | 4.8380      | 1.0000     | 42,357.87     |

### 5.2. Solutions of the Multi-Objective Optimization Problem

It is well known that all points in the Pareto front correspond to optimal solution for design variables. In Figures 4 and 5, the upper-right point corresponds to the optimal solution if power output $f_1$ is set as a unique objective, while the lower-left point corresponds to the optimal design variables if the mass $f_2$ of the PEH is set as a unique objective. Any other point in the Pareto front provides an intermediate solution that balances the posed objectives $f_1$ and $f_2$.

Table 5 shows a few solution values together with their corresponding parameter values obtained by the present approach. The rightmost column of the table indicates the algorithm that provides the solution to the BPF.

**Table 5.** Some solutions belonging in optimal Pareto front.

| Power     | Mass    | $L^*$   | $H^*$   | R         | Algorithm |
|-----------|---------|---------|---------|-----------|-----------|
| 283.46553 | 0.53482 | 0.62928 | 0.34647 | 310.01575 | NSGA-III  |
| 284.36259 | 0.53595 | 0.62587 | 0.35000 | 325.36746 | NSGA-II   |
| 310.98901 | 0.66365 | 0.99147 | 0.33902 | 63.05137  | GDE3      |
| 320.36665 | 0.67724 | 0.99596 | 0.35000 | 69.58982  | GDE3      |
| 322.27086 | 0.67842 | 0.99905 | 0.35000 | 76.01130  | GDE3      |

As mentioned above, GDE3 achieves solutions that generate a power output greater than that generated by NSGA-II and NSGA-III. As it may be seen, these maximum values for power output correspond to a length of piezoelectric layers which cover the whole harvester as well as to lower values of electrical resistance.

Our experiments showed that the imposed constraint on bending stresses seem to have no effect on the solutions provided by the optimization algorithms. Similar solutions are obtained by excluding the stress constraints altogether. Nevertheless, for different problem parameters (e.g., type of the piezoelectric material, geometric dimensions of the structure, forcing term etc.) the proposed model can guarantee that the solutions provided will always respect the stress limits of the manufactured energy harvesting system. Table 6 contains some solutions of the optimal Pareto front obtained in [14] where an additional constraint on the mass of the system is imposed, namely $f_2 \leq 0.3267$. It can be seen from that table, that the greater values for output power are obtained when the mass approaches the upper limit of the imposed constraint.

**Table 6.** Some solutions belonging to the optimal Pareto front of [14].

| Power | Mass | $L^*$ | $H^*$ | R | Algorithm |
|---|---|---|---|---|---|
| 165.71184 | 0.32666 | 0.24316 | 0.11184 | 139.61471 | NSGA-II |
| 165.57695 | 0.32656 | 0.22207 | 0.12203 | 160.53703 | NSGA-III |
| 165.27248 | 0.32619 | 0.20690 | 0.12934 | 190.74535 | GDE3 |

A comparison of Table 5 to Table 6 shows that the values of the power outputs obtained by the present approach are greater than that of [14]. However, due to the conflict between mass function and power output function, these values correspond to greater values for the mass. The multi-objective optimization procedure and the provided multiple Pareto-optimal solutions give the designer the opportunity to make a better decision in selecting one final optimal solution depending on the particular implementation requirements.

## 6. Conclusions

A multicriteria design optimization problem has been studied in this work in order to achieve an optimal design of cantilever PEH. The harvester is considered as a bimorph plate structure with two embedded piezoceramic layers in series connection. A finite element model has been and verified with experimental results found in the literature. The objectives taken into consideration are maximization of power output and minimization of mass system. Maximum bending stress is considered as an additional constraint in the design optimization. Three state-of-the-art multi-objective constrained optimization algorithms (NSGA-II, NSGA-III and GDE3) have been applied optimize the geometric dimensions the geometric dimensions and the electrical component of the PEH. Numerical results show that all these algorithms start to converge to the base Pareto optimal front around the 10th generation. Nevertheless, only GDE3 manages to extend to solutions generating power greater than about 284 mW/g$^2$. Performance of the three multi-objective optimization algorithms (has been assessed by calculating several metrics demonstrating the good quality of the solutions. Overall, the results show that GDE3 achieves solutions that generate higher maximum power output and performs best according to MPFE, MS and HV metrics, compared to the two other algorithms.

**Author Contributions:** Conceptualization, G.F. and C.G.; methodology, G.F. and N.A.; software, G.F., C.G., N.A. and A.M.; validation, C.G. and A.M.; formal analysis, C.G.; investigation, C.G. and A.M.; resources, C.G. and N.A.; data curation, C.G.; writing—original draft preparation, C.G. and N.A.; writing—review and editing, G.F., C.G., N.A. and A.M.; visualization, C.G. and A.M.; supervision, G.F. All authors have read and agreed to the published version of the manuscript.

**Funding:** This research received no external funding.

**Institutional Review Board Statement:** Not applicable.

**Informed Consent Statement:** Not applicable.

**Data Availability Statement:** New data were created and analyzed in this study. Data sharing not applicable.

**Acknowledgments:** Special thanks are due to the developers of the "PlatEMO" software, the developers of the python module "pfevaluator",the developers of the python module "OApackage" and the developers of the "WebPlotDigitizer v4.5" software.

**Conflicts of Interest:** The authors declare no conflict of interest.

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
