# Peer review of "Multicriteria Approach for Design Optimization of Lightweight Piezoelectric Energy Harvesters Subjected to Stress Constraints"

_information, doi:10.3390/info13040182_

Round 1
Reviewer 1 Report
Thanks for giving me this opportunity to review this paper. The manuscript is about optimization models to design cantilever PEH with embedded piezoceramics.
The Authors did a valuable work. The manuscript is well composed, written, and nicely discussed. In my opinion, the manuscript is suitable for publication in information journal after the completion of minor revision.
-Author just discussed about piezoceramics which they are brittle, recently there are many piezo composites with high output power. I recommend author to discus about these pizocomposite in their model and their difference with piezoceramics. Some of these piezoceramics presented in the following papers:
-Wearable Electronic Textiles from Nanostructured Piezoelectric Fibers. Adv. Mater. Technol. 2020, 5, 1900900
-"Mechanical Energy Harvesting and Specific Potential Distribution of a Flexible Piezoelectric Nanogenerator Based on 2-D BaTiO3-Oriented Polycrystals." ACS Sustainable Chemistry & Engineering (2022)
